# Fostering Pre-Professionals and Learning Experiences in End-of-Life Care Contexts: Music Therapy Internship Supervision

**DOI:** 10.3390/healthcare12040459

**Published:** 2024-02-11

**Authors:** Amy Clements-Cortés, Sara Klinck

**Affiliations:** 1Faculty of Music, The University of Toronto, Toronto, ON M5B 1W8, Canada; 2Faculty of Music, Wilfrid Laurier University, Waterloo, ON N2L 3C5, Canada; sklinck@wlu.ca

**Keywords:** supervision, palliative care, internships, music therapy, end-of-life, interdisciplinary collaboration

## Abstract

Certified music therapists use music within therapeutic relationships to address human needs, health, and well-being with a variety of populations. Palliative care and music therapy are holistic and diverse fields, adapting to unique issues within end-of-life contexts. Palliative care music therapy has been formally practiced since the late 1970s and affords a variety of benefits, including pain and anxiety reduction, enhancement of quality of life, emotional expression, and relationship completion. The training of music therapists varies around the globe, but clinical supervisors play a key role in skill acquisition. Clinical supervisors support pre-professionals as they realize the application of their training, foundational competencies, and authentic therapeutic approaches in end-of-life care, while navigating the challenges and rewards of this work. This article is a narrative review which offers background information on palliative care music therapy, and reports the authors’ viewpoints and reflections on supervision strategies and models employed with music therapy interns in palliative care settings based on their experiences. Approaches are shared on supporting pre-professionals as they begin working in palliative care contexts, as well as implications for supervision practices.

## 1. Introduction

A palliative care approach aims to improve quality of life and mitigate suffering for those navigating terminal or life-limiting illnesses. Palliative care occurs in various contexts: in-home support, inpatient palliative care units within hospital settings, long-term care facilities, retirement residences, residential hospices, community hospices, and community agency programs. Hospices typically also offer day wellness programs for those negotiating life-threatening or life-limiting illness, and bereavement care for those grieving the death of their loved one.

Music and music therapy experiences are becoming more common in end-of-life care contexts [1,2,3,4]. Music therapists are allied healthcare professionals who provide music therapy experiences to persons at end of life, often in collaboration with the interdisciplinary team [3]. “Music Therapists (MTAs) use music purposefully within therapeutic relationships to support development, health, and well-being. Music therapists use music safely and ethically to address human needs within cognitive, communicative, emotional, musical, physical, social, and spiritual domains” ([5], para #1). There are many music opportunities in end-of-life care, ranging from entertainment to recreational music to music therapy. Each experience has value but is also different. For an experience to be considered music therapy, four elements are needed: a music therapist, client with a need, music, and a therapeutic relationship [6]. See Figure 1.

Clements-Cortés and Bartel [7] share a four-level model for mechanisms of response in music therapy which includes learned cognitive responses, cognitive activation, stimulated neural coherence, and cellular genetic responses. Researchers have also highlighted the scientific basis for standardized therapeutic music experiences clustered into the sensorimotor, speech/language, and cognitive domains [8]. Meta-analyses and systematic reviews offer scientific evidence that music therapy can support a wide range of symptom improvements, including pain and anxiety [9,10,11,12,13,14]. Further research and descriptive articles point to the efficacy of music therapy for relationship completion [15], emotional expression [16], and improvements in heart rate, blood pressure, augmented relaxation, and wellness [4]. Similarly, McConnell and Porter [17] highlight enhanced physical comfort, increased emotional well-being, enhanced social interaction, and improved spiritual well-being as valuable outcomes of music therapy at end of life.

Palliative care and music therapy are both holistic and diverse fields, continually adapting to unique issues in these end-of-life contexts. Palliative care music therapy aims to improve quality of life while supporting goals such as symptom management, emotion regulation, communication, and spiritual expression [4]. Clements-Cortés [18] found that both live and recorded music provided in music therapy treatment resulted in statistically significant reductions in pain perception and the enhancement of physical comfort.

The thematic analysis of palliative care music therapy research and practice highlights that music therapy addresses physical, psychosocial, and whole-person care [19]. These experiences may include receptive and active interventions such as singing, playing instruments, songwriting, clinical improvisation, and guided relaxation, based on evidence and best practices. For example, Vesel and Dave [20] note the benefit of music therapy for supporting pain management, increasing energy and quality of life while decreasing anxiety. Similarly, Gallagher et al. [21] also found music therapy assisted with anxiety and stress reduction as well as perceived pain level, overall quality of life, mood improvement, and acceptance of death.

Reidy and MacDonald [22] note: “Ongoing barriers to music therapy include the challenge to obtain adequate sustainable funding and the misunderstanding of music therapy as entertainment” (p. 1605). Funding for music therapy services varies considerably depending on the country, province, state, and location of care. In hospice or inpatient palliative care settings, if there is a music therapist on the interdisciplinary team the patient would not necessarily pay for music therapy, as it would be covered by their fees to be in that setting, insurance, or the government. However, for example, only 6% of hospice programs employ music therapists [20]. It is typical that a patient would pay for the service if they were receiving palliative care in the community in Canada. Customarily, fees for the clinical supervision of music therapy pre-professionals’ internships in palliative care are either covered by university training programs (when considered a placement course), and/or by the organization that employs the music therapist (on staff or contract) if clinical supervision is included in the job responsibilities.

Training to be a therapist and professional caregiver for those who are dying is a rewarding but challenging process. Clinical supervisors strive to support pre-professionals as they apply their training, grow foundational competencies, and develop research-informed therapeutic approaches and resources. The requirements to serve as a clinical music therapy supervisor vary depending on where a music therapist is practicing, but typically involve the completion of a set number of direct client contact hours, extensive work experience, and course work or training in supervision. Where the authors reside and practice as registered music psychotherapists, we adhere to the requirements as outlined by the College of Registered Psychotherapists of Ontario wherein the supervisor must adhere to the following:Be a member in good standing of a regulatory college;Have five years’ extensive clinical experience;Complete 1000 direct client contact hours and 150 h of clinical supervision;Provide a signed declaration noting they understand CRPO’s definitions of clinical supervision, clinical supervisor, and the scope of practice of psychotherapy [23].

To date, very little has been written specifically on the topic of clinical supervision in palliative care music therapy. This limits the ability to conduct a thorough scoping review yielding significant insights in this area of practice. Therefore, as a starting point, this narrative review, report, and viewpoint article aims to highlight significant areas for consideration when supporting pre-professionals’ learning experiences and providing clinical supervision. Music therapy internships are first described in general, followed by the process of launching into, preparing for, and debriefing these internships in palliative care contexts. Supervision models and approaches are implicated as the authors share from the relevant and supporting existing literature, with a focus on their own lived experiences as clinicians and clinical supervisors in palliative music therapy and end-of-life care. Recommendations are made for future research, projects, and the advancement of clinical supervision in this area of practice.

## 2. Music Therapy Internships

The education requirements to practice as a music therapist vary throughout the globe, but in Canada and the United States, music therapists are required to have a minimum of a bachelor’s degree in music therapy and to become certified by the Certification Board for Music Therapists [24]. Students take a variety of courses including psychology, music, and music therapy research, while also partaking in practical music therapy placements [25].

Music therapy internships are valuable clinical experiences fostering pre-professionals’ improvement in the clinical, musical, and personal skills essential for practice [26]. Music therapy training places an emphasis on practice and internship as authentic settings linking academic training with professional clinical work [27]. Pre-professionals receive clinical supervision as they respond to the needs of clients, communities, and healthcare systems, and learn the role of the therapist in these spaces. Supervised clinical training programs at the pre-professional level are required for certification or accreditation in the field of practice [28], and entry-level competencies must be demonstrated and evaluated by the end of internship.

All of the locations for palliative care provision are potential placement sites for pre-professionals, and there are unique learning experiences offered within each context. These comprehensive internship and supervision processes aim to train student therapists to reflect and integrate therapeutic, musical, and academic learning in the creation of their music therapist identity [29]. While these are clearly formative seasons of intensive learning, Clements-Cortés [27] highlights that there is much variability in internship supervision practices. Contexts are diverse and inconsistencies arise in size of caseload, clinical settings, frequency of supervision, and observed practice. Pre-professionals’ learning experiences are also impacted by previous clinical training, interaction with the interdisciplinary team, and the overall supervisory approach.

Generalized learning focuses on demonstrating foundational and working knowledge relevant to clinical contexts and developing musical, clinical, and professional competencies and capacities to empathically understand clients. Further, self-awareness of one’s own presence and influence on the state of clients, on therapeutic relationships, or the music therapy process, and how one uses musical, nonmusical and verbal techniques to communicate and interact with clients, team members, and healthcare systems are also emphasized. Additionally, pre-professionals learn how to work collaboratively and disseminate music therapy work to other professionals and interdisciplinary teams [29]. Clinical supervisors evaluate pre-professionals’ progress and competence in the aforementioned areas of learning, skill, and knowledge, throughout internship. In Canada, standardized evaluation forms are completed at the end of internship, highlighting strengths and ongoing or future learning goals. All direct clinical and non-clinical internship hours (totaling a minimum of 1000 h) are documented alongside recommendations for writing the board certification exam [24,30] and acquiring accreditation/certification with the national association [25].

Studies highlighting the perceived experiences of music therapy internships in Canada [27] and the United States [26] point to some common challenges faced by pre-professionals. Feelings of unpreparedness when starting internship, concerns regarding finances, not measuring up to expectations, lack of verbal counseling skills, and difficulty putting theory into practice were all expressed [27,31,32,33]. Clinical, musical, and personal skill development were notably increased during internship, and being immersed in clinical work fostered skill building, improved musicianship, reduced anxiety, and increased confidence [26].

More specific concerns were voiced by students placed in hospice and palliative care settings over developing emotional attachment to patients who are at end of life, anxiety and uncertainty regarding their potential reactions to certain situations including the death of a patient, how to navigate brief therapy that may be provided over just one session, and how to learn a relevant music repertoire quickly [27,31,33]. By better understanding the aims, concerns, and issues faced by pre-professionals, internship supervisors, educators, and care team members can be more responsive and better equipped to support pre-professionals in their learning experiences and skill building throughout internship.

## 3. Launching into Palliative Care Music Therapy Internship: Preparation and Debriefing

The following section predominantly highlights our lived experiences supporting and supervising music therapy pre-professionals in various end-of-life contexts. It has been our observation that pre-professionals begin an internship with varied theoretical knowledge and understanding of palliative care, highly dependent on whether or not their instructors had experience and interest in this area of practice. There are currently no full courses centered on palliative music therapy alone in Canadian training programs, and few in other international training programs, so pre-professionals rely on supervised placements to grow their skills and expand their hands-on knowledge.

When pre-professionals begin working in end-of-life care, there are many questions that arise that might not surface if the individual was working with a different demographic or population. For example, young pre-professionals are now encountering death on a daily basis and may not have experienced death to any significant degree in their own lives. Many pre-professionals are navigating other developmental and life-stage personal experiences and challenges such as buying a house, getting married, or starting a family, and unless someone in their circle is going through a death-related experience, death is likely not at the forefront of their daily lives. Pessin et al. [34] maintain providing end-of-life care evokes existential concerns and varies considerably in healthcare practitioners. When pre-professionals begin working in palliative care, transpersonal and existential questions can rise and cause additional anxieties on top of those already present with respect to navigating a new environment and attempting to bring the knowledge learned in the classroom to the workplace and clinical setting. In order to prepare pre-professionals to undertake death-related work, clinical supervisors will recommend readings and encourage review of the literature that is relevant and interesting in the fields of end-of-life care, bereavement, and grief theory. Pitts and Cevasco [33] also found that music therapy pre-professionals placed in hospice and palliative care felt they lacked musical preparedness, expressing a desire for further guitar training and an expansion of their musical repertoire for this context. Therefore, tangible and practical music repertoires are also shared to help pre-professionals prepare resources that can be utilized and adapted to meet client preferences or needs in the moment.

Initial supervisory conversations highlight a growing understanding of the palliative care approach, a care-versus-cure focus, symptom management, and enhancing quality of life. Concepts of burnout and compassion fatigue are discussed alongside self-care strategies for coping with highly emotional circumstances. Pre-professionals are encouraged to be aware of personal experiences with death and loss, and how these may potentially impact the way they approach the work. Supervisors can prepare resources and worksheets to help pre-professionals chart their personal loss history or to reflect on their experiences and responses in different seasons of loss. Themes can then be brought up in supervision if the student wishes to discuss them further.

In the first few weeks of placement, having the opportunity to observe or shadow an experienced music therapist is helpful for understanding how to approach the work in this context and build confidence before facilitating sessions independently. A frequently asked question we have each heard from our supervisees is: “What can I expect when I walk into the room of a person who is dying?” To help pre-professionals with this, it is beneficial to discuss death and dying openly and ask them to reflect on their own thoughts and experiences with death. This can bring up philosophical questions as well as transpersonal reflections, which can be helpful. It is also supportive to share logistical information to prepare pre-professionals for things they might see, like rapid breathing, changes in skin colour (mottling), rapid decline, etc. While this information might be more commonplace in other allied health fields’ training programs, we must remember that the training of music therapists involves them working with individuals across the lifespan, so there is not always time to focus on each population in great depth. Some pre-professionals specialize in working with a demographic, but that typically comes in the later stages of school, post graduation, or concurrently with a clinical placement. Therefore, they may not have as much foundational knowledge on some of the common terminology or issues that will be present in end-of-life care.

We task our pre-professionals to consider the purpose of their interventions and articulate to themselves when forming goals and objectives why they are there. We encourage them to think reflectively about how their practice differs from what an entertainer does. For example, music therapists conduct assessments, write goals and objectives, and respond to the client in the moment. They use their training to foster change and support the client in a holistic approach. When this discussion ensues, it is also beneficial to discuss the guidelines for professional practice and scope of practice [30,35] and link them into a discussion on the core competencies they are implementing while working towards their board certification as music therapists.

In Canadian palliative care contexts, it is common to have either on-site or off-site supervisors. The intern–supervisor relationship is complex, and is affected by parallel processes, styles of supervision, transference, and countertransference [36]. This alliance greatly impacts a pre-professional’s clinical practice and sense of self-worth, enhancing opportunities for professional growth as they integrate theory and research with practice and application in various end-of-life settings. Whether on site or off site, our experience has been that regular supervisory check-ins are essential for pre-professionals to feel adequately supported with the aforementioned questions and concerns alongside the clinical work and debriefing. The interdisciplinary team can play an important on-site role in these cases. Pre-professionals have expressed lacking confidence in forming and maintaining healthy peer-working relationships or communicating clearly with other staff [27]. However, attending rounds, care meetings, and having regular formal or informal touch-bases with other professionals can help pre-professionals feel less isolated and more connected and accepted as a part of the overall palliative care team. Developing a strong interprofessional network of peers is especially vital if the music therapy supervisor themselves is in an off-site role.

## 4. Potential Supervision Models

Clinical supervisors supporting pre-professionals in palliative care might consider adopting characteristics from various music-centered, competency-based, developmental, student-centered, and psychodynamic models of supervision.

Music improvisation and the review of video/audio clips are two ways that music-centered supervision [37] could be utilized during supervision. Music improvisation can be engaged to oscillate freely between verbal and musical processes. Improvising as the client helps supervisees access emotions and develop further empathy for the client’s situation and responses. Pedersen et al. [29] describe the use of creative media in group supervision. This could take the form of (1) students drawing or writing poetry while listening to another student’s clinical description and then giving feedback; (2) musical role-playing of client and therapist; or (3) students improvising on their sensations and impressions of the client. Reviewing video/audio clips together in supervision can help develop the practice of clinical listening, response, and interpretation of the musicking [37], while recalling the student’s embodied experiences of the session [29].

To further practice and develop applicable skills, competency-based supervision [38,39] is grounded in guidelines and is fundamental at the pre-professional training level and during internship. Typically, the supervisor will model an approach, methods, and techniques, and the supervisee is focused on learning skills and specific behaviours. Structured learning modules can then be used for competency building within three stages of internship: the beginning, the middle, and the final.

Similarly, ref. [29] recommend three levels for supervising student music therapists (SMTs), inspired by an expanded version of the Integrated Developmental Model of Supervision (IDM) [40]. These levels help supervisors and supervisees identify developmental stages in the internship and advance our understanding of how to best support pre-professionals throughout the process. In level 1, students are concerned with acquiring and applying the correct tools and techniques, and are focused on their own performance and approval or disapproval from patients and supervisors. The supervisor’s role is to decrease anxiety, encourage self-awareness, and build competency. In level 2, students demonstrate an increased understanding of clients’ perspectives. This intense focus on the client may result in strong countertransference reactions with reduced self-awareness. The role of the supervisor is to witness, be aware of, and support all reactions, offering advice and ideas flexibly as the student finds a meaningful way forward. The supervisor can also clarify transference/countertransference patterns and emphasize the importance of oscillating attention and awareness between the self and the client [29]. At the third level, pre-professionals accept the strengths and weaknesses of the self and of the client, and demonstrate competent awareness, consciousness, and disclosure of their countertransference reactions alongside well-developed empathy. The supervisor remains supportive and the supervision itself becomes increasingly mutual. Pre-professionals aim to balance meeting the needs of the client while developing an approach and experiences that feel authentic to them. Progress and process in this model present as linear or circular movement, moving back and forth between levels in students’ understanding and identification of themselves as future professional music therapists [29]. See Figure 2 for the authors’ depiction of this model as it relates to palliative care music therapy internships.

These stages and levels can be balanced with student-led supervision, which encourages growth in pre-professionals’ self-awareness and the ability to articulate their needs. Student-centered, or person-centered, approaches to supervision [38,41] are interested in understanding the supervisee’s feelings, thoughts, and reactions to the clients from their own perspective. Authenticity and genuineness are emphasized for both parties, and the supervisor should hold unconditional positive regard for the uniqueness of each supervisee’s perspectives. Essential is the belief that pre-professionals have the skills, knowledge, and capacity to resolve personal and therapeutic issues.

The psychodynamic model of supervision [29,38,42] is an advanced form used with graduate students and professionals. Here, supervision is considered an inter-subjective context where both unconscious and conscious mental processes are created and developed [29]. Emphasis is placed on helping students grow self-awareness and understand unconscious processes such as countertransference, parallel processes, defenses, and resistance, and how these manifest and can be worked with in both the clinical/therapeutic and supervisory relationships [38,42,43]. The aim is to increase insight regarding the impact of past events and interpersonal dynamics on current ones in order to identify patterns of relating, feeling, and thinking.

The aforementioned models should be appraised, applied, and adapted to fit the needs of the supervisee and the supervisor’s own authentic approach. Implications for palliative care supervision should further take into consideration population- and context-specific clinical issues, challenges, and needs.

## 5. Implications for Music Therapy Supervision Approaches in Palliative Care

Supervision sessions can take many forms and draw from several verbal, musical, and embodied approaches. Modeling and co-facilitating palliative music therapy sessions can provide pre-professionals with rich opportunities to observe and shadow experienced therapists, ask questions, and learn in a multi-faceted way. Experiential workshopping during supervision gives space for practicing skills and role-playing scenarios. Role play can lead to shifting perspectives and a more holistic understanding of a client or a situation [29]. This might include practicing meeting patients for the first time, navigating family dynamics, and verbal counseling at end of life. Music-centered workshopping and the demonstration of techniques such as entrainment to match and support breathing; lyric or vocal improvisation; spontaneous and structured songwriting; assisted instrumental improvisation; song transformation or extemporization; and practicing flow between verbal and music elements of the session can grow understanding and hands-on comfort in applying these resources in palliative care music therapy practice. By reflecting through embodied experiences such as role play, music, or other artistic media in supervisory processes, pre-professionals increase self- and other-awareness, enhance cognitive capacities, and grow an understanding of unconscious dynamics [29].

Supervisors seek to support pre-professionals’ questions and anxieties surrounding what the therapeutic process looks like. Overall, Atkin et al. [44] note that access to psychological support at end of life is often limited. Given that music therapists might only be on site in the location where end-of-life services are provided on a limited basis (i.e., one time per week) and that patient condition can change rapidly from day to day, pre-professionals may have less opportunity to have a set number of sessions with clients. This can create apprehension for pre-professionals and also challenges the typical planning of the treatment process. It is important to help them realize it is okay to pause and always ask the client what they need when they are unsure and to gain comfort being in the space of truly listening and responding. This can facilitate the essential learning of “being with” as opposed to “doing to/for” clients. O’Callaghan [45] notes “As patients’ conditions fluctuate, sessions may be flexibly scheduled, range from minutes to over 1 h, and can be offered occasionally to almost daily.” (p. 214). Pre-scheduled individual sessions are not typical in inpatient hospice or palliative care contexts, so pre-professionals must frequently navigate an unpredictable workday. Instead of focused advanced preparation for specific clients, pre-professionals are encouraged to complete an initial check-in with the interdisciplinary team and with clients/families when they first arrive for each placement day. This helps gather important updates about changing conditions, recent deaths, and which clients might be (self-) referred or prioritized for music therapy that day. Pre-professionals can then prepare themselves and gather resources for sessions with more intention and knowledge of present needs.

A beneficial approach to offer pre-professionals is to meet the client prepared to actively listen and gather information on what they would like or need support in order to determine the focus for the session. Supervisors should also coach pre-professionals to hone their ability to observe and assess various functional, cognitive, emotional, relational domains, and environmental factors in the first minutes of each session. As pre-professionals learn to pay attention to the client’s expressed wishes and their own observations, combined with recommendations from the interdisciplinary team, they grow a more fulsome understanding of the unique experiences of dying and the reality that care needs may change drastically one day to the next. It is vital all therapists practice cultural responsiveness in order to provide holistic care [46] and learn about what death and dying mean to their client. This is particularly important in the scope of music therapy given that therapists might be using specific music related to rituals, and also to ensure we are using music that is permissible and appropriate. At times, the client might have the energy and desire to work on projects or treatment objectives started previously, and this is then the pathway. There is additional supervisory encouragement and focus given to preparing and practicing how to respond to clients’ preferred music experiences with immediacy, since time and number of sessions may be limited, and decision making may have to be expedited.

Internships in end-of-life care can understandably feel emotionally heavy, as pre-professionals are regularly navigating death and dying with their clients. Supervisors can recommend self-reflection practices [31,32,47] for debriefing and processing such as journaling; experimenting with mindfulness, meditation, and gratitude; planning and scheduling intentional time to process during the day/week; and creating honouring, remembrance, or closure rituals to acknowledge and express their own grief after a client dies. These rituals might take the form of writing a letter or song to the deceased, or creative expression through poetry, musical improvisation, or singing/playing a piece of music that was meaningful for that client and your work together. Encouraging pre-professionals to establish cognitive–emotional boundaries between placement and home can help develop an awareness of leaving the clinical work behind at the end of the day while cultivating healthy patterns for work–life balance.

Various forms of supervision should be valued and accessed to deepen processing, consider multiple perspectives, and hear from diverse voices on clinical issues. Individual supervision can allow for specific reflections on caseload, alliance building, session challenges, and clinical or ethical considerations. Personal challenges, countertransference, and issues with the therapeutic relationship can be shared and explored in a more secure space [29]. If pre-professionals are co-supervised between professional colleagues, they have access to potentially diverse or complimentary clinical approaches, different vantage points on clinical issues, and a wider array of resources or opportunities to workshop and reflect on session experiences. In this collaborative method [42], supervisors share the responsibility of supporting pre-professionals’ development and ethical practice, while benefiting from opportunities for peer debriefing and problem solving together in order to support pre-professionals most effectively.

Supervisor-facilitated group supervision can provide avenues for exploring techniques and sharing frustrations and insecurities with peers, which in turn can help validate and normalize the experience while launching new understandings and insights. The role of the group supervisor is to lead the sharing and feedback experience, ensuring that the debrief between peers is positive and appropriate, and to point out countertransference issues to help pre-professionals recognize and understand complex processes [29]. Pre- or new professionals can engage further social support, motivation, and encouragement through peer-run supervision [48]. These peer-led groups can provide an avenue to ask and receive both professional and personal advice regarding clinical work, professional opportunities and challenges, how to balance work and self care, and offer a shared experience of transitional student–professional identities.

## 6. Future Research and Projects to Advance Clinical Supervision in End-of-Life Care

This article offers perspectives and viewpoints on supervision based on the authors’ experiences. There are a number of opportunities for research to advance clinical supervision practices in end-of-life care. Here are a few ideas:Interview music therapy interns on their perceived training needs and experiences with supervision approaches.Survey nursing, medical, counseling, art therapy, and other trainees working in palliative care on their training and supervision experiences in order to identify unique issues and common challenges across disciplines.Develop supervision protocols that outline number of sessions, activities/structure for each session type, training techniques, and underlying theoretical bases.Pilot culturally adapted versions of manualized supervision protocols with representative minority populations and compare skills/knowledge development to standard approaches. A mixed-methods study examining the impact of individual vs. group vs. peer supervision formats on competency acquisition and retention after internship would be beneficial.An analysis of pre-professionals’ pre-internship self-evaluations and learning goals at the start of internship by tracking progress quantitatively on those indicators, along with supervisor evaluations over 6–12 months.

## 7. Conclusions

Regular supervision is essential for pre-professionals’ feelings of preparedness and for debriefing the work in palliative care. The supervisory relationship actively unfolds in ways that can leave each party transformed by the questions, challenges, and growth that occur throughout the internship [49]. When pre-professionals feel adequately supported in their learning, concerns, applied practice, and accomplishments in palliative care placements, they experience how the unique challenges and rewards of this work can help grow their overall clinical, musical, and professional competencies, and shape their authentic music therapy approach while valuing interdisciplinary collaboration.

Supervisors are encouraged to anticipate challenges and proactively provide spaces to discuss these and offer resources. Key areas of challenges and opportunities for supervisors to provide support are in fostering interprofessional collaborations, scaffolding competency acquisition, promoting discussions and reflections on anxiety, grief, loss, and death, the formation of boundaries, self-care, and comfort in working in diverse spaces.

By extension, these practices would be beneficial to other allied healthcare professionals, as well as nursing and physicians in training. Interprofessional peer support networks are encouraged to foster a greater understanding of the scope of practice as well as an increased understanding of how to meet the needs of patients and provide quality end-of-life care. Future research is needed on understanding pre-professionals’ experiences in navigating end-of-life care internships to subsequently foster an increased knowledge of how supervisors can support them.

## Figures and Tables

**Figure 1 healthcare-12-00459-f001:**
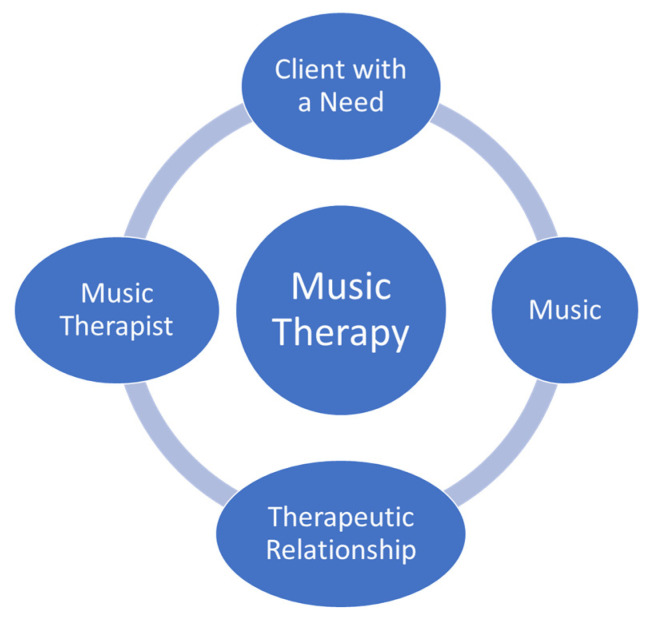
Elements of the music therapy experience.

**Figure 2 healthcare-12-00459-f002:**
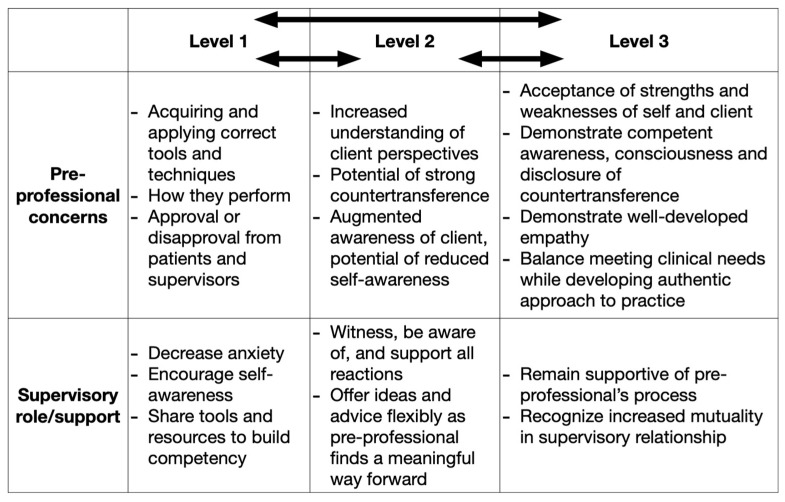
Summary of levels for supervising pre-professionals inspired by IDM.

## Data Availability

Data are contained within the article.

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
