# Peer review of "Fostering Pre-Professionals and Learning Experiences in End-of-Life Care Contexts: Music Therapy Internship Supervision"

_healthcare, 2024, doi:10.3390/healthcare12040459_

Round 1

Reviewer 1 Report (Previous Reviewer 2)

Comments and Suggestions for Authors

Fine job addressing the points raised by this editor and those of the other editors. Your writing is clear. and your additions of graphic depictions only enhances what was already a strong article, that you definitively positioned as non-research. 

Author Response

Thank you very much for confirming the changes meet your approval.

Reviewer 2 Report (Previous Reviewer 3)

Comments and Suggestions for Authors

Dear Authors, thank you for allowing me to review this second version of your manuscript.

I found it considerably improved and I appreciated you had considered my previous suggestions. However, it seems that the methodology you have used relies only on your personal knowledge about the topic, rather than a process of literature review. I would suggest to consider disclosing this as a limit of the study.

Moreover, my perception is that the focus on supervisors is sometimes missing in favour of other aspects of music therapy, try to be consistent throughout the text with the aim of this study.

Author Response

Dear Authors, thank you for allowing me to review this second version of your manuscript.

I found it considerably improved and I appreciated you had considered my previous suggestions. However, it seems that the methodology you have used relies only on your personal knowledge about the topic, rather than a process of literature review. I would suggest to consider disclosing this as a limit of the study.

Please see our changes in Teal highlights. The yellow highlights are from the first round of review.

This is not a study. We have tried to make that even clearer this time by adding text at the end of the abstract and also part 1. We have added that not conducting a scoping review is a limitation.

Moreover, my perception is that the focus on supervisors is sometimes missing in favour of other aspects of music therapy, try to be consistent throughout the text with the aim of this study.

Thank you for your comment. We are trying to respond to all reviewer comments which are quite different. We have added a little more on music therapy’s scientific basis to respond to reviewer 4 in the introduction in this second revision. We feel the background information on palliative care music therapy is important to the paper and was asked for in the first submission of our paper. We also feel the majority of the paper is still on supervision but the other information is essential to the understanding of supervision issues.

Reviewer 3 Report (Previous Reviewer 4)

Comments and Suggestions for Authors

All my comments have been addressed

Author Response

Thank you for confirming the changes we have made meet your approval. 

Reviewer 4 Report (Previous Reviewer 5)

Comments and Suggestions for Authors

Reviewing a manuscript entitled, “Fostering Pre-Professionals and Learning Experiences in End-of-Life Care Contexts: Music Therapy Internship Supervision” by Clements-Cortés A and Klinck S, this is an article focusing on music therapists training method. Although the authors responded to many of my concerns in 1st revision, I don't think this manuscript is an article but review or report because this does not include any research conducted by the authors.

 The revised text in yellow marker still gives a narrative impression. The authors should provide the scientific basis for music therapy in palliative care. For example, You mentioned “enhanced physical comfort, increased emotional well being.”, you should provide scientific evidence regarding these matters.

Comments on the Quality of English Language

Minor editing is required. But there are no major problems. 

Author Response

Reviewing a manuscript entitled, “Fostering Pre-Professionals and Learning Experiences in End-of-Life Care Contexts: Music Therapy Internship Supervision” by Clements-Cortés A and Klinck S, this is an article focusing on music therapists training method. Although the authors responded to many of my concerns in 1st revision, I don't think this manuscript is an article but review or report because this does not include any research conducted by the authors.

Please see our changes in Teal highlights. The yellow highlights are from the first round of review.

This is correct. The paper is not a research study and we made that clearer in the abstract and introductory paragraph noting this was not a research study. We have added the additional words report in the abstract to solidify this for the reader

“This article offers background information on palliative care music therapy, and provides a report on the status of music therapy supervision by sharing the authors’ viewpoints and reflections discussing supervision strategies and models for music therapy interns in palliative care settings based on their experiences.”

Additionally, a sentence has been added to the end of section 1 to make this clearer.

 The revised text in yellow marker still gives a narrative impression. The authors should provide the scientific basis for music therapy in palliative care. For example, You mentioned “enhanced physical comfort, increased emotional well being.”, you should provide scientific evidence regarding these matters.

The outcomes you highlighted are listed by the authors cited in that paragraph (McConnell & Porter, 2017). The sentence order has been changed to clarify that the evidence was shown in  their article.

It seems beyond the scope of our paper to go into further detail on this. Reviewer 2 also noted that we are starting to get away from the focus of the paper which is supervision if we add more on palliative care music therapy. Therefore we have added three citations and briefly made note of the scientific support.

Round 2

Reviewer 4 Report (Previous Reviewer 5)

Comments and Suggestions for Authors

This is acceptable quality in the content. However, I am not convinced that the paper category at line 1 is article. Please check with the editorial office whenever possible.

Author Response

Thank you for confirming the changes are acceptable. We agree that line 1 should indicate this is a review not article according to the manuscript types for this journal. 

This manuscript is a resubmission of an earlier submission. The following is a list of the peer review reports and author responses from that submission.

Round 1

Reviewer 1 Report

Comments and Suggestions for Authors

Dear authors, thank you for a chance to read interesting your paper.

Abstract:

Authors should indicate what type of article it is. That I will not find research there.

Introduction:

Line 28 - unnecessary space.

At the end there should be a clear statement of the article's purpose and what readers can expect to find in the subsequent sections. 

Next Parts:

Line 123 - unnecessary space.

Authors should consider incorporating charts or figures to better describe what is in their text. The article has potential for visuals.

Conclusion:

No information on potential future research.

Reviewer 2 Report

Comments and Suggestions for Authors

Very well-written report on an essential topic. Good clarity overall, but the sentence beginning on line 105 is dense, and it took several times reading through it for it become clear. Perhaps broken into several sentences would allow for a more pleasing flow. 

The inclusion of group supervision is great, but you could consider mentioning peer-run vs supervisory facilitated group supervision here. 

Reviewer 3 Report

Comments and Suggestions for Authors

Dear authors, thank you for allowing me to review this interesting manuscript describing barriers and strategies to promote the learning experiences of music therapists.

Although I found the manuscript well-written and focused on a topic of clear interest for healthcare providers, there are some major concerns that I have identified and limiting the methodological quality of your report. I summarise below the major concerns for your consideration.

It is not clear what do you intended with "pre-professionals"; you jumped several times along the text between the terms "students" and "pre-professionals" without a clear definition of what is your population of interest. At first, there is a long description of pre-professionals intnerships (Page 2, Lines 70-74), then the students were introduced for the first time (Page 2, Lines 77-84), I could imagine you are referring to the same population, isn't it?

In the same way, there is a fundamental lack of the aim of this study, that led to another important concern: what is this paper? A series of recommendations? A literature review? A position statement? A quality improvement proposal? 

Consequently, the methods are totally lacking. Thus, it is unclear how the cited articles were retrieved and analysed (search strategy, inclusion/exclusion criteria, selection procedure, analysis). 

Finally, there are some parts along the main four chapters in which the manuscript has been subdivided that were not sustained by a citation, making the considerations you provided as resulting from a series of anecdotical recommendations. There are paragraphs of more than 10 lines that totally lack references; you should address this issue before providing recommendations on how to educate future professionals on the basis of your personal perspective.

It is a pity because, for example, I have sincerely appreciated the fourth section (potential supervision models), but is the only one that has a structure that is credible at a scientific readers' eye.

I hope my considerations will help you improve the quality of your study and design more methodological-underpinned studies in the future. 

Reviewer 4 Report

Comments and Suggestions for Authors

This article discusses strategies for supervising music therapy interns in palliative care settings. It highlights the challenges pre-professionals can face when starting internships in end-of-life contexts, including lack of preparation, developing emotional attachments to dying patients, anxiety regarding death, and learning skills quickly. The authors recommend ways supervisors can support students, such as assigning preparatory readings, providing musical resources, discussing palliative care concepts, and addressing self-care.

The article then outlines supervision approaches that can be used to foster pre-professionals' development, including music-centered, competency-based, developmental, student-centered, and psychodynamic models. Implications for supervision in palliative care are shared, emphasizing the need to support interprofessional collaboration, competency building, discussions of grief/anxiety, boundary formation, and self-reflection. Supervisors are encouraged to respond to patients' fluctuating needs and provide both group and individual supervision to process the intense emotional components of this work. Overall, tailored supervision strategies can help pre-professionals gain clinical and personal skills essential for professional music therapy practice in end-of-life care.

The article is a viewpoint discussing supervision strategies and models for music therapy interns in palliative care settings, rather than presenting an original research study. However, here are some suggestions to enhance this area of research in the future

  1. Conduct controlled intervention studies comparing a specific supervision model to standard training. Include quantitative skills assessments and knowledge tests before and after to evaluate effectiveness.
  2. Interview music therapy interns on their perceived training needs and experiences with supervision approaches using an open-ended questionnaire. Code responses to identify recurring themes and areas for improvement.
  3. Survey nursing, medical, counseling, art therapy and other trainees working in palliative care on their training and supervision experiences. Identify common challenges across disciplines as well as unique issues.
  4. Develop manualized supervision protocols that outline number of sessions, activities/structure for each session type, supervisor qualifications, training techniques, underlying theoretical basis. Test protocol fidelity through supervisor reporting metrics.
  5. Examine impacts of individual vs group vs peer supervision formats on competency acquisition and retention after internship completion using mixed methods.
  6. Have interns conduct self-evaluations and set learning goals at the start of internship. Track progress quantitatively on those indicators along with supervisor evaluations over 6-12 months.
  7. Follow interns into their professional roles and evaluate workplace preparedness through self-reports and employer evaluations at periodic intervals up to 5 years post-graduation. Gather data on career trajectories, sense of burnout and job satisfaction.
  8. Pilot culturally-adapted versions of manualized supervision protocols with representative minority populations and compare skill/knowledge development to standard approaches. Refine supports to optimize outcomes.

Reviewer 5 Report

Comments and Suggestions for Authors

Reviewing a manuscript entitled, “Fostering Pre-Professionals and Learning Experiences in End-of-Life Care Contexts: Music Therapy Internship Supervision” by Clements-Cortés A and Klinck S, this is an article focusing on music therapists training method. However, it is not the authors' original work, but describes the training methods used so far. At least to me it looks like a report, not an article, and everything is almost narrative and there is no basis for what has been described.

 What qualifications does the clinical supervisor mentioned in the abstract have?

 In the introduction section, the authors mentioned “Music and music therapy experiences are becoming more common in end-of-life care contexts.”. The authors should include multiple citations if they are more general.

 In the introduction section, the authors mentioned “For an experience to be considered music therapy, four elements are needed: music therapist, client with a need, music, and the therapeutic relationship (Bradt et al., 2013).”. The authors should provide an illustration or figure of how the four elements fit together.

 The authors should explain a relationship between the medical insurance system and music therapy system, even if the example is from the United States or Canada. If it is medical care, will the patients pay for the treatment?

 It is necessary to cite manuscripts that shows that music therapy is effective for the patients.

 An illustration of music therapy internships is needed. What steps do students take during their internship? Are there any guidelines or standards to meet for music therapy internships?

 What kind of position is a clinical supervisor? Are you a doctor on the medical team? What qualifications does your supervisor have? Are there any guidelines regarding clinical supervisors?

 The authors should attach an illustration of three levels for supervising student music therapists.

 The relationship between music therapy internships and music therapy supervision is unclear. After completing the internship, what are the steps to get a job in the field and become a supervisor?

Comments on the Quality of English Language

Minor editing is required. But there are no major problems.